# Study of Orofacial Function in Preschool Children Born Prematurely

**DOI:** 10.3390/children9030360

**Published:** 2022-03-04

**Authors:** Mei-Chen Chang, Hsiu-Yueh Liu, Shun-Te Huang, Hsiu-Lin Chen

**Affiliations:** 1Department of Nursing, Kaohsiung Veterans General Hospital, Kaohsiung City 813414, Taiwan; yn20030824@gmail.com; 2Department of Oral Hygiene, College of Dental Medicine, Kaohsiung Medical University, Kaohsiung City 807378, Taiwan; hyliu@kmu.edu.tw (H.-Y.L.); shunteh@kmu.edu.tw (S.-T.H.); 3Division of Pediatric Dentistry and Special Care Dentistry, Department of Dentistry, Kaohsiung Medical University Hospital, Kaohsiung City 807378, Taiwan; 4Department of Pediatrics, Kaohsiung Medical University Hospital, Kaohsiung City 807378, Taiwan; 5Department of Respiratory Therapy, College of Medicine, Kaohsiung Medical University, Kaohsiung City 807378, Taiwan

**Keywords:** preterm, children, oral facial features, NOT-S, chewing and swallowing

## Abstract

Children born prematurely often exhibit orofacial dysfunction. We conducted Nordic Orofacial Test Screening and analyzed chewing and swallowing functions of 243 children aged 3–5 years, consisting of 142 and 101 children born full-term and preterm, respectively, to evaluate the orofacial function of preschool premature children. Categorical variables were analyzed using chi-square test for a comparison. The univariate analysis of variance was used to analyze the effects of birth weight, gestational age, intubation at birth, use of nasal continuous positive airway pressure support after birth, and use of nasogastric tube on the chewing and swallowing functions of children born prematurely. In this survey, term-born children had a higher incidence of bad oral habits, grinding teeth while sleeping, and abnormal gulping compared to preterm-born children. Preterm-born children had a higher incidence of choking, decreased mouth opening (<30 mm), abnormal dental arch form, abnormal palatal vault, and dysarthria compared to term-born children.

## 1. Introduction

The World Health Organization defines preterm birth as live birth before completion of 37 weeks of gestation. Preterm infants can be further subdivided on the basis of weight at birth: <2500 g, low birth weight; <1500 g, very low birth weight (VLBW); and <1000 g, extremely low birth weight (ELBW). Preterm birth is the leading cause of death worldwide for children <5 years old [1]. The initial days of life of premature infants are challenging compared with those of full-term infants. Numerous reports have shown that many survivors of preterm birth have a lifetime of disability, including learning disabilities, visual and hearing impairments, language and speech impairments, attention deficits and hyperactivity, poor social skills, stunting, and functional delays [2,3,4,5]. However, reports regarding the orofacial function of children born prematurely are limited. Orofacial function results from complex activities of the central nervous and neuromuscular systems [6]. It includes a multitude of vital actions, such as sucking, breathing, chewing, swallowing, obtaining nutrition, talking, coughing, snoring, vomiting, communicating through emotions, facial expression, and social interaction [7,8]. Orofacial dysfunction in a child causes distress to the family, particularly orofacial dysfunction causing communication and nutritional difficulties [9]. Ingestion is the basic function for survival, and it is closely related to orofacial function. During the first few years of life, the neural system matures, and the orofacial skeletal structures rapidly grow [10]. In preterm infants, this development is affected, impairing their suck–swallow mechanism. Therefore, feeding is done nonorally in most preterm infants, hindering their orofacial development. The nonoral feeding process continues to develop in the later stages of childhood, but this condition has not been studied in depth [11,12].

The aim of this study was to evaluate the orofacial and swallowing–chewing functions in preschool children born prematurely.

## 2. Materials and Methods

This cross-sectional study was conducted from September 2018 to August 2019. The children enrolled in this study were 3–5 years old and were from two medical centers (Kaohsiung Medical University Hospital and Kaohsiung Veterans General Hospital) in Kaohsiung City, Taiwan. The procedures, contents, and importance of the survey and questionnaire were explained to parents by neonatologists, and written consent was obtained from 243 parents who agreed to enroll their children in the study.

The exclusion criteria were participants with intellectual disability, mental developmental delay, cerebral palsy, systemic disorders, or obvious orofacial dysfunction or nonavailability of data.

This study was approved by the IRB of Kaohsiung Medical University Hospital (Protocol number: KMUHIRB-20140107) and the Kaohsiung Veterans General Hospital (Protocol number: VGHKS15-CT3-11).

We used Nordic Orofacial Test-Screening (NOT-S) to evaluate orofacial dysfunction in this study. The NOT-S was developed by Bakke et al. in 2007 to assess orofacial dysfunction and identify individuals who need further evaluation or treatment for oral-motor function [7,13]. The NOT-S can be used in children >3 years old, adolescents, and adults, and medical personnel with different occupational background can easily use it after introduction and calibration [13]. It has been translated into several languages, including Chinese, and it is verified to have high reliability and sensitivity [7]. During the face-to-face interview, parents assisted their children to answer the questions. The duration of NOT-S ranged from 5 to 7 min. The Chinese version of NOT-S was validated with high reliability; the sensitivity and specificity values of the screening were 0.96 and 0.63, respectively [7]. We conducted a pilot study on five participants before this study. The test–retest reliability of the NOT-S was assessed; the Spearman’s rank correlation coefficient was 0.801, which revealed high reliability. NOT-S contains 12 domains in the form of a structured interview and clinical examination. The interview consists of six domains: (I) sensory function, (II) breathing, (III) habits, (IV) chewing and swallowing, (V) drooling, and (VI) dryness of the mouth; examination consists of six domains: (1) the face at rest, (2) nasal breathing, (3) facial expression, (4) masticatory muscle and jaw function, (5) oral-motor function, and (6) speech. Each domain contains one to five items. If the answer to one of the questions or the performance of one of the tasks meets the criterion for impaired function, “yes” is recorded for that item. Any “yes” in a domain earns 1 point, thus indicating a dysfunction in the scored domain. The maximum score is 12 points. High scores indicate increased oral dysfunction [7].

We used a structured questionnaire to assess chewing and swallowing functions in children (Table 1). The questionnaire contained four domains: (I) basic information, (II) daily eating condition, (III) oral habits, and (IV) chewing and swallowing functions. These domains were selected based on factors that affect chewing and swallowing functions. Furthermore, expert validity and reliability of the questionnaire were examined. For expert validity, questionnaire validity was tested by six experts with relevant knowledge regarding and experience in questionnaire applicability. The index of Content Validity values were between 88.02% and 100%, and the validity was 0.96. For questionnaire reliability, a pretest was conducted on five primary caregivers of the young children. The test–retest reliability analysis was performed, and the correlation was 100% with internal consistency.

The IBM SPSS 22.0 (IBM Corp., Armonk, NY, USA) software package was used for statistical analysis, and the significance level was established at *p* < 0.05. Categorical variables were analyzed using chi-square test for a comparison of proportions. Univariate analysis of variance was used for analyzing the association of chewing and swallowing functions with birth weight, weeks of birth, number of days with intubation, number of days with nasal continuous positive airway pressure (NCPAP) support, and number of days with nasogastric (NG) tube insertion.

## 3. Results

In total, 243 children aged 3–5 years (101 preterm-born and 142 term-born children) were enrolled, with 65 and 178 children with birth weight < 1501 and >1500 g, respectively.

Table 2 presents the current nutritional status of enrolled children. Pureed soft foods were introduced at 4.0–4.1 months of corrected age to children born preterm and with birth weight < 1501 g, but they were introduced at 4.8–5.1 months to term-born children and children with birth weight > 1500 g. Preterm-born children and children with birth weight < 1501 g were started on pureed soft foods earlier than were term-born children and children with birth weight > 1500 g. The preterm-born children had lower percentiles of weight, height, and body mass index (BMI) of the growth curve than did term-born children at the enrolled time. VLBW children had lower percentiles of weight and BMI of the growth curve than did children with birth weight > 1500 g.

Table 3 presents the conditions at birth for VLBW children. We divided these VLBW children into two groups based on birth weight into >1000 and <1001 g (ELBW). Children with ELBW had significantly lower Apgar scores at 1 and 5 min and longer intubation days, NCPAP days, NG tube placement days, and hospitalization days than those with >1000 g birth weight (*p* < 0.05). These results indicate poorer conditions after birth in ELBW preterm children than in preterm children with birth weight between 1001 and 1500 g.

Table 4 presents the orofacial conditions and chewing and swallowing functions of enrolled preschool children. We observed that bad oral habits, grinding teeth while sleeping, and abnormal gulping were lower but choking while swallowing water, abnormal mouth opening (<30 mm), abnormal dental arch form, abnormal palatal vault, and dysarthria were higher in preterm children than in term children (*p* < 0.05). Furthermore, the same situations were found except the abnormal occlusive force was higher in the birth weight with <1501 g, while we divided the groups by birth weight with <1501 g and >1501 g.

Table 5 presents the mean gestational age and mean birth weight of enrolled children with abnormal and normal orofacial conditions and chewing and swallowing functions. The NOT-S results showed that children with higher mean gestational age and mean birth weight presented with bad oral habits and teeth grinding during sleep. The assessment of chewing and swallowing functions revealed that children with lower mean gestational age and birth weight presented with choking while swallowing water, abnormal mouth opening (<30 mm), abnormal occlusal force, abnormal dental arch form, abnormal palatal vault, and dysarthria. However, abnormal gulping was observed in children with a high mean gestational age and mean birth weight.

We further analyzed the association of orofacial conditions and chewing and swallowing functions with the management of VLBW preschool children while in the neonatal intensive care unit (Table 6). The orofacial conditions and chewing and swallowing functions of VLBW preschool children were not related to intubation days while in the neonatal intensive care unit. However, longer NCPAP days were significantly related to the presence of breathing-snoring during sleep and abnormal mouth opening (<30 mm); longer days of NG tube placement were significantly related to the presence of abnormal mouth opening (<30 mm), conscious cough, and modified water swallowing test (MWST) in VLBW preschool children.

## 4. Discussion

Our study revealed that because of the introduction to pureed soft foods before 4 months of age, the percentiles of the growth curve and BMI curve were poorer in 3–5 years old premature children than in full-term children. A significant number of studies have suggested that in preterm infants, solid foods are introduced even before 4 months because preterm infants are at a greater risk of developing feeding or nutritional disorders, and they may have lower physical growth compared with full-term infants [14,15]. At 3 years of age, preterm children weighed significantly less than full-term children did [15]. Pridham et al. revealed that feeding skills of preterm infants varied widely in terms of sucking and eating semisolid and solid food [12]. Inadequate feeding capabilities in preterm infants often lead to poor nutrition, growth failure, and pathologies, although pureed soft food has been introduced earlier in premature children than in term children [14,16]. The incorporation of supplementary feeding was started earlier and was more difficult in preterm infants than in full-term infants, and delays in feeding development and difficulties in transitioning to new textures and tastes were common in preterm infants [17]. The causes of these difficulties in preterm infants may be their immaturity, extended hospitalization, chronic illnesses, prolonged exposure to unpleasant oral tactile experiences, or neurological impairments [18].

Our study revealed that term-born children and children with a birth weight of >1500 g had a higher incidence of bad oral habits, grinding teeth while sleeping, and abnormal gulping than did preterm-born children and children with birth weight <1501 g. Bad oral habits are common in children, which include nonnutritive sucking habits (thumb, finger, pacifier, or tongue), lip biting, and bruxism (teeth grinding) events. These conditions are associated with anger, hunger, sleep, tooth eruption, and fear [19]. According to a previous study, sucking habits were only related to parents’ education, and child feeding methods were not influenced by sex, birth status, or family income [20]. The habit of teeth grinding was the most prevalent in children 2–4 years old, but this was not associated with the gestational age and birth weight according to the study of Ferrini et al. in 2008 [21]. These findings were different from those of our current study.

Our study reported that preterm infants were introduced to solid foods earlier than full-term infants were. A significant number of studies have suggested that gulping appeared earlier in preterm infants than in full-term infants, and that preterm infants were introduced to solid foods at an earlier age than full-term infants were. Munching appeared earlier in preterm infants than in full-term infants, but preterm infants learned to chew slower than did full-term infants [22,23]. Early practice and frequent “teaching” may facilitate the development of oral feeding coordination [24]. The coordination of sucking, swallowing, and respiring is required for safe feeding and the prevention of aspiration and impaired respiratory status. Most of the full-term infants are born with developed feeding skills, but preterm infants might experience aspiration while swallowing during respiration [25], which was also revealed in our study. The mouth breadth, bite force, dental arch form, and palatal vault height have been found to increase with age [26,27]. Our findings revealed higher rates in abnormal presentations in terms of mouth width, bite force, dental arch shape, and dome height of the oral cavity in preterm-born children than in term-born children of comparable ages.

Studies have revealed that no significant difference exists in the number of occurrences of speech disorders between preterm low-birth-weight children and term-born children, but some reports have shown a significant delay in speech sound acquisition in preterm children [28,29]. In our study, we found significantly higher occurrences of dysarthria in preterm low-birth-weight children than in term-born children. This might be due to delay in the maturation of speech sound acquisition as mentioned in previous reports.

In the present study, we also found that children with birth weight <1001 g had poorer birth conditions than those with birth weight between 1001 and 1500 g, including lower Apgar score, more endotracheal tube placement days, more NCPAP days, and longer hospitalization days. Preterm infants born with ELBW (birth weight <1000 g) often stay in neonatal intensive care and undergo many medical procedures, such as suctioning from the airway, tube feeding, and even intubation, which may have negative impacts on oral-sensory and oral-motor function [12,30]. Studies have revealed that the NG or orogastric tubes for gavage feeding may lead to a risk of altered oral sensitivity; facial defensiveness; delayed oral feeding; immature jaw movement in biting and chewing; underdeveloped functions of swallowing semi- solids particles; poor coordination of sucking, swallowing, and breathing; delay in the transition from drinking from a bottle to drinking from a cup; prolonged duration of mealtimes; and reduced amount of food eaten at meals [12,31]. Furthermore, the findings of the current study showed long days of NG tube placement during hospitalization after birth had negative effects on mouth opening, conscious cough, and MWST in VLBW preschool children. A study suggested that low-birth-weight infants who undergo orotracheal intubation were at a risk of poor sucking ability at term and at 3 months of corrected age [32]. The relationship between intubation duration and the development of oral feeding skills in premature infants is unclear [33]. Our study revealed that intubation time did not affect oral facial function, chewing, and swallowing. This might be due to a short intubation period in our enrolled children. Ferrara et al. demonstrated that NCPAP alters the pharyngeal swallowing mechanism [21] and tracheal aspiration event [34] in neonates. The prolonged intubation duration was associated with low gestational age and then increased likelihood of oral feeding initiation while on continuous positive airway pressure among these premature infants who displayed a poor coordination of suck and swallow [33]. The other study revealed that premature infants at high risk of sleep-disordered breathing (reported to snore ≥3 days/week) had gastroesophageal reflux and a family history of snoring [35]. Our study showed that the condition of breathing-snoring while sleeping was associated with an increased number of NCPAP days in VLBW premature born children. Nasal prongs used for the interface of NCPAP may irritate the mucosa of the nostrils, inducing the swelling of the mucosa of the nose later. This issue warrants further investigation. The long-term use of intubation, NCPAP, and tube feeding leads to oral sensory and oral–motor dysfunction and difficulty in mouth opening in the premature infants, which warrant further investigation.

There are no neonatal treatment guidelines that take the growth and disabilities of the oral and craniofacial region into account now. Based on the results of our study, the efforts to decrease the days of NCPAP and NG placement for premature infants are necessary to improve the outcome of the orofacial function. Oral feeding should be introduced early for premature infants to decrease the days of NG tubes. However, this is often delayed due to the need for prolonged NCPAP. In addition, most caregivers are worried that NCPAP would disrupt sucking–swallowing–breathing coordination and then result in tracheal aspiration [36]. The optimal strategy for weaning very preterm infants from NCPAP during hospitalization is still unclear. However, one randomized controlled trial of weaning strategies for preterm infants on NCPAP reported that using a high flow nasal cannula might reduce the duration of NCPAP, which might be a way to decrease the days of NCPAP [37].

As our study was a cross-sectional study, not a longitudinal study, it is impossible to clarify if the dysfunction will improve with age. A longitudinal study of adolescents aged 12–14 and 17–19 from a population of Swedish adolescents born preterm revealed a poor oral health-related quality of life, especially for orthodontic treatment. The good dentist–patient relationship should be emphasized in preterm-born children to improve oral health [38,39].

## 5. Conclusions

In conclusion, poor birth conditions and the use of a feeding tube and NCPAP in premature infants are associated with an increased incidence of choking, abnormal mouth opening (<30 mm), abnormal dental arch form, abnormal palatal vault, and dysarthria. Furthermore, premature children with the condition of breathing-snoring while sleeping had longer days of NCPAP. Premature children with abnormal mouth opening had longer days of NCPAP and NG placement compared to term children. Moreover, the longer days of NG placement affect the conditions of abnormal results for conscious cough and MWST. The efforts to decrease the days of NCPAP and NG placement for premature infants are necessary to improve the outcome of orofacial function.

## Figures and Tables

**Table 1 children-09-00360-t001:** Assessment of chewing and swallowing functions of children.

Variable	Characteristics	Categories
Gulping	Swallowing food after chewing	
	Swallowing food without chewing	
Choke (water and liquid classes)	Non-choke: No coughing during eating	
	Choke: Coughing more than 3 times during eating	
The amount of open mouth	21–30 mm	
	31 mm or more	
Occlusal force	Normal: Can chew ordinary diet	
	Abnormal: Can only chew soft food	
Open bite	≦2 mm	
	>4 mm	
Dental arch form	Circular shape	
	Non-Circular shape	
Palate vault	Normal Palate	
	Abnormal Palate	
Dysarthria	Normal: clear articulation of phonemes	
	Abnormal: poor articulation of phonemes	
Conscious cough	Good: Strong contraction of abdominal muscles, exercise accompanied by cough sounds	
	Poor: Abdominal muscle contraction, coughing sound without accompanying movement	
Modified Water Swallowing Test (MWST)	Normal: 3 mL water, swallow more than 3 times within 30 s	
	Abnormal: 3 mL of water, swallowed less than 2 times within 30 s	

**Table 2 children-09-00360-t002:** Current nutritional status of preschool children.

Variable	GA < 37 Weeks	GA ≥ 37 Weeks	*p*-Value	BBW < 1501 g	BBW > 1500 g	*p*-Value
	(*N* = 101)	(*N* = 142)		(*N* = 65)	(*N* = 178)	
	Mean ± SD	Mean ± SD		Mean ± SD	Mean ± SD	
Average age of the first time eating pureed food (Month)						
Fruits	4.0 ± 1.3	5.1 ± 0.9	<0.001	4.1 ± 1.3	4.8 ± 1.1	<0.001
Minced-Toast	6.5 ± 2.5	7.5 ± 2.1	<0.001	6.7 ± 2.9	7.2 ± 2.1	0.025
Vegetable	8.0 ± 2.8	9.4 ± 2.2	<0.001	8.1 ± 3.1	9.1 ± 2.3	0.006
Meat	10.2 ± 3.6	11.0 ± 2.0	<0.001	10.3 ± 4.2	10.8 ± 2.1	0.006
Current status of development						
Weight growth curve (%)	33.4 ± 30.2	46.0 ± 31.9	0.002	33.0 ± 30.4	45.0 ± 31.3	0.003
Weight growth curve < 3% ^1^	14 (13.9)	10 (7.0)	0.079	12 (18.5)	12 (6.7)	0.007
Height growth curve (%)	33.3 ± 28.7	47.8 ± 31.5	<0.001	36.8 ± 32.8	42.2 ± 31.5	0.161
Height growth curve < 3% ^1^	23 (22.8)	15 (10.6)	0.010	14 (21.5)	24 (13.5)	0.126
BMI growth curve (%)	42.7 ± 32.0	52.4 ± 33.5	0.033	40.0 ± 31.8	51.4 ± 33.6	0.018
BMI growth curve < 3% ^1^	12 (11.9)	14 (9.9)	0.615	11 (16.9)	15 (8.4)	0.058

^1^ Presented as n (%). GA: gestational age; BBW: birth body weight; BMI: body mass index.

**Table 3 children-09-00360-t003:** Conditions after the birth of very-low-birth-weight premature infants.

Variable	Total	BBW 1001–1500 g	BBW < 1001 g	*p*-Value ^1^
	(*N* = 65)	(*N* = 50)	(*N* = 15)	
	Mean ± SD	Mean ± SD	Mean ± SD	
Apgar 1 min	5.9 ± 1.6	6.2 ± 1.3	5.1 ± 1.9	0.043
Apgar 5 min	7.8 ± 1.2	7.9 ± 1.3	7.2 ± 1.0	0.011
Day of intubation	12.7 ± 19.7	6.5 ± 15.0	26.7 ± 22.7	<0.001
Day of NCPAP	26.7 ± 19.7	22.9 ± 17.9	39.5 ± 20.4	0.005
Day of NG tube placement	51.9 ± 31.0	43.6 ± 24.1	79.6 ± 35.8	<0.001
Day of hospitalization	71.5 ± 333.8	60.5 ± 26.7	108.2 ± 29.3	<0.001

NCPAP: nasal continuous positive airway pressure; NG: nasogastric; BBW: birth body weight. ^1^ Comparison based on the Mann–Whitney test between 1001–1500 g and <1001 g.

**Table 4 children-09-00360-t004:** Orofacial conditions and chewing and swallowing functions of enrolled preschool children.

Variable	GA < 37 Weeks	GA ≥ 37 Weeks	*p*-Value	BBW < 1501 g	BBW > 1500 g	*p*-Value
	(*N* = 101)	(*N* = 142)		(*N* = 65)	(*N* = 178)	
	*N* (%)	*N* (%)		*N* (%)	*N* (%)	
NOT-S						
Breathing-snoring while sleeping	12 (11.9)	20 (14.1)	0.617	9 (13.8)	23 (12.9)	0.850
Habits (oral habits)	36 (35.6)	76 (53.5)	0.006	21 (32.3)	91 (51.1)	0.009
Grinding teeth while sleeping	15 (14.9)	38 (26.8)	0.027	7 (10.8)	46 (25.8)	0.012
Abnormal chewing and swallowing	61 (60.4)	97 (68.3)	0.202	39 (60.0)	119 (66.9)	0.321
Speech- Pronunciation is not standard	22 (21.8)	19 (13.4)	0.085	16 (24.6)	25 (14.0)	0.051
Assessment of chewing and swallowing in children						
Abnormal gulping	19 (18.8)	46 (32.4)	0.018	10 (15.4)	55 (30.9)	0.016
Choke	12 (11.9)	5 (3.5)	0.012	11 (16.9)	6 (3.4)	<0.001
The abnormal amount of open mouth	6 (5.9)	0 (0.0)	0.003	6 (9.2)	0 (0.0)	<0.001
Abnormal occlusal force	43 (42.6)	51 (35.9)	0.294	34 (52.3)	60 (33.7)	0.008
Abnormal open bite	6 (5.9)	5 (3.5)	0.371	4 (6.2)	7 (3.9)	0.461
Abnormal dental arch form	29 (28.7)	21 (14.8)	0.008	24 (36.9)	26 (14.6)	<0.001
Abnormal palate vault	25 (24.8)	14 (9.9)	0.002	21 (32.3)	18 (10.1)	<0.001
Dysarthria	21 (20.8)	15 (10.6)	0.027	16 (24.6)	20 (11.2)	0.009
Abnormal conscious cough	26 (25.7)	49 (34.5)	0.145	13 (20.0)	62 (34.8)	0.270
Abnormal MWST	34 (33.7)	49 (34.5)	0.891	19 (29.2)	64 (36.6)	0.328

GA: gestational age; BBW: birth body weight; NOT-S: Nordic Orofacial Test-Screening; MWST: Modified Water Swallowing Test.

**Table 5 children-09-00360-t005:** Sample distribution of orofacial and chewing and swallowing domains according to birth week and birth weight.

Variable	Gestational Age (Weeks)		*p* -Value	Birth Weight (g)		*p* -Value
	Negative	Positive		Negative	Positive	
	Mean ± SD	Mean ± SD		Mean ± SD	Mean ± SD	
NOT-S						
Breathing-snoring while sleeping	35.9 ± 4.1 (211)	35.6 ± 4.9 (32)	0.742	2505.2 ± 890.2 (211)	2530.3 ± 957.0 (32)	0.883
Bad oral habits	35.3 ± 4.3 (131)	36.5 ± 4.0 (112)	0.018	2364.2 ± 899.1 (131)	2677.4 ± 869.0 (112)	0.006
Grinding teeth during sleep	35.7 ± 4.3 (190)	36.7 ± 3.5 (53)	0.044	2437.0 ± 922.5 (190)	2765.2 ± 53.6 (53)	0.018
Chewing and swallowing	35.5 ± 4.2 (85)	36.1 ± 4.2 (158)	0.311	2473.8 ± 912.2 (85)	2527.2 ± 891.5 (158)	0.659
Speech—Pronunciation is not standard	36.1 ± 4.0 (202)	34.8 ± 4.9 (41)	0.065	2558.8 ± 856.5 (202)	2260.7 ± 1053.2 (41)	0.052
The assessment chewing and swallowing in children						
Gulping	35.5 ± 4.3 (178)	36.9 ± 3.7 (65)	<0.017	2407.0 ± 909.8 (178)	2786.7 ± 804.8 (65)	0.003
Choke	36.0 ± 4.1 (226)	32.8 ± 4.5 (17)	<0.001	2556.9 ± 875.8 (226)	1865.6 ± 959.1 (17)	0.002
The abnormal amount of open mouth	36.0 ± 4.0 (237)	28.2 ± 2.3 (6)	<0.001	2544.8 ± 877.5 (237)	1077.5 ± 318.2 (6)	<0.001
Occlusal force	36.3 ± 3.8 (149)	35.2 ± 4.7 (94)	0.057	2602.4 ± 818.1 (149)	2359 ± 996.7 (94)	0.040
Open bite	35.9 ± 4.1 (232)	34.7 ± 4.8 (11)	0.361	2522.6 ± 895.8 (232)	2211.7 ± 919.0 (11)	0.262
Dental arch form	36.3 ± 3.9 (193)	34.1 ± 4.8 (50)	0.001	2610.6 ± 847.4 (193)	2114.4 ± 981.3 (50)	<0.001
Palate vault	36.3 ± 4.0 (204)	33.6 ± 4.5 (39)	<0.001	2603.9 ± 840.7 (204)	2009.7 ± 1023.7 (39)	<0.001
Dysarthria	36.1 ± 4.0 (207)	34.3 ± 5.1 (36)	0.014	2572.5 ± 860.4 (207)	2140.5 ± 1022.9 (36)	0.007
Conscious cough	35.5 ± 4.3 (168)	36.7 ± 3.9 (75)	0.041	2442.8 ± 937.2 (168)	2140.5 ± 1022.9 (75)	0.092
MWST	35.7 ± 4.2 (160)	36.1 ± 4.0 (83)	0.511	2466.3 ± 931.3 (160)	2655.5 ± 787.3 (83)	0.593

NOT-S: Nordic Orofacial Test-Screening. MWST: Modified Water Swallowing Test. gestational age and birth weight presented as mean ± SD; number presented in parentheses.

**Table 6 children-09-00360-t006:** Univariate analysis of variance for intubation, CPAP, NG, and chewing and swallowing in very-low-birth-weight children.

Variable	Day of Intubation (*N* = 65)		*p*-Value	Day of NCPAP (*N* = 65)		*p*-Value	Day of NG Tube Placement (*N* = 65)		*p*-Value
	Negative	Positive		Negative	Positive		Negative	Positive	
	Mean ± SD	Mean ± SD		Mean ± SD	Mean ± SD		Mean ± SD	Mean ± SD	
NOT-S									
Breathing-snoring while sleeping	11.4 ± 17.4	21.3 ± 29.6	0.158	24.7 ± 18.1	39.6 ± 25.0	0.034	51.1 ± 32.7	53.7 ± 27.7	0.755
Bad oral habits	10.2 ± 17.4	13.1 ± 22.1	0.556	25.3 ± 20.3	29.7 ± 18.4	0.401	25.3 ± 20.3	29.7 ± 18.4	0.401
Grinding teeth during sleep	11.4 ± 19.9	9.0 ± 9.1	0.756	26.6 ± 20.2	28.0 ± 15.6	0.857	51.4 ± 32.2	56.4 ± 19.7	0.686
Chewing and swallowing	9.8 ± 17.3	12.0 ± 20.1	0.66	26.4 ± 20.3	26.9 ± 19.5	0.911	47.3 ± 23.7	55.0 ± 35.0	0.332
Speech—Pronunciation is not standard	9.4 ± 16.0	16.4 ± 25.8	0.203	26.6 ± 20.2	27.1 ± 18.5	0.937	53.0 ± 30.9	48.7 ± 32.2	0.636
The assessment chewing and swallowing in children									
Gulping	10.4 ± 16.3	14.9 ± 30.5	0.497	25.5 ± 18.1	33.4 ± 27.0	0.246	52.0 ± 31.1	51.5 ± 32.1	0.964
Choke	10.8 ± 18.8	15.6 ± 22.5	0.586	26.1 ± 19.5	34.8 ± 22.4	0.343	51.5 ± 31.1	57.4 ± 32.9	0.683
The abnormal amount of open mouth	10.6 ± 18.8	16.3 ± 22.5	0.483	24.7 ± 19.3	45.0 ± 4.0	0.016	49.1 ± 30.1	79.8 ± 22.2	0.019
Occlusal force	7.0 ± 14.4	14.9 ± 21.8	0.091	23.7 ± 18.5	29.5 ± 20.5	0.241	45.5 ± 21.7	57.836.9	0.109
Open bite	11.3 ± 19.2	8.3 ± 15.2	0.757	27.2 ± 20.0	19.3 ± 11.6	0.437	52.9 ± 31.4	37.3 ± 21.0	0.333
Dental arch form	12.5 ± 21.4	8.8 ± 13.8	0.451	26.7 ± 21.8	26.8 ± 15.7	0.983	50.7 ± 34.4	54.0 ± 24.6	0.666
Palate vault	10.9 ± 19.8	11.7 ± 17.4	0.874	28.0 ± 20.1	24.0 ± 16.9	0.437	49.5 ± 27.6	56.9 ± 37.4	0.378
Dysarthria	9.4 ± 16.0	16.4 ± 25.8	0.203	26.6 ± 20.2	27.1 ± 18.5	0.937	53.0 ± 30.8	48.7 ± 32.2	0.636
Conscious cough	11.1 ± 19.5	11.2 ± 17.1	0.955	24.5 ± 18.3	35.8 ± 23.1	0.063	46.5 ± 25.1	73.5 ± 42.6	0.004
MWST	11.6 ± 20.3	9.8 ± 15.2	0.718	24.7 ± 17.8	31.7 ± 23.3	0.193	46.8 ± 25.4	64.2 ± 39.6	0.039

NOT-S: Nordic Orofacial Test-Screening. NCPAP: nasal continuous positive airway pressure. NG: nasogastric tube. MWST: Modified Water Swallowing Test.

## Data Availability

The data presented in this study is available on request from the corresponding author.

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
