# Peer review of "Study of Orofacial Function in Preschool Children Born Prematurely"

_children, 2022, doi:10.3390/children9030360_

Round 1

Reviewer 1 Report

I only suggest to add a paragraph, in the Materials and Methods section, regarding the institutional Ethics Committee's authorization for performing the study.

Author Response

Thank you for your suggestion. We have added a paragraph in the Materials and Methods section (page 2, lines 67-69) regarding the institutional Ethics Committee's authorization for performing the study.

Reviewer 2 Report

  I appreciate the opportunity to review an article entitled “Study of Orofacial Function in Preschool Children Born Prematurely”. This report showed oral dysfunctions in 3~5 year-old children born prematurely. It’s a interesting topic for pediatricians and dentists. I think that they should know about those dysfunction after premature babies survive the neonatal period. I will list some points I have noticed at this time.  Hope these help.

1).The numbers in Table 5 and 6 are misaligned. They should be fixed.

2).Is it possible that the dysfunctions shown in your study will improve with age? Please discuss this point with some references.

3).It is easy to see that some treatment, such as NCPAP or NG tube, during the neonatal period could have a significant impact on the subsequent growth of the children. In a sense, some of those oral dysfunctions you pointed out could be considered iatrogenic, although everyone knows that these treatments are necessary to save the lives of children. Are there any the neonatal treatment guidelines that take the growth and disabilities of oral and craniofacial region into account? If so, please list them.

Author Response

Thank you for your review and valuable suggestions. The responses to your suggestions are as follows.

1) We have revised Table 5 and Table 6 for optimizing the layout.

2) Because our study was a cross-sectional study, not a longitudinal study, it is impossible to clarify if the dysfunction will improve with age. A longitudinal study of adolescents aged 12–14 and 17–19 from a population of Swedish adolescents born preterm revealed a poor oral health-related quality of life, especially for orthodontic treatment. The good dentist-patient relationship should be emphasized in preterm-born children to improve oral health [38,39].

We have added this description in the Discussion section.

3) There are no neonatal treatment guidelines that take the growth and disabilities of the oral and craniofacial region into account now. Based on the results of our study, the efforts to decrease the days of NCPAP and NG placement for premature infants are necessary to improve the outcome of the orofacial function. Oral feeding should be introduced early for premature infants to decrease the days of NG tubes. However, this is often delayed due to the need for prolonged NCPAP. In addition, most caregivers are worried that NCPAP would disrupt sucking–swallowing–breathing coordination and then result in tracheal aspiration [36]. It is still unclear the optimal strategy for weaning very preterm infants from NCPAP during hospitalization. However, one randomized controlled trial of weaning strategies for preterm infants on NCPAP reported that using a high flow nasal cannula might reduce the duration of NCPAP, which might be a way to decrease the days of NCPAP [37].

We have added this description in the Discussion section.

Reviewer 3 Report

Dear authors,

congratulations to your clinical highly relevant study on n = 243 children and their oral function status.

The reference list of the manuscript contains 35 titles, and is without inappropriate self-citations. The manuscript is clear, with a good rate of novelty and significance. The manuscript present scientific resound and the design appropriate to test the hypothesis. The methods and software are clear described, with sufficient details to permit another researcher to reproduce the results. All aspects regarding the figures/tables/images are appropriate, and they are easy to interpret and understand. The presentation and the analyzed date are written in proper way. The presentation of the results are at high standard, with appropriate statistics. The results offer a development in the present knowledge, are significant, and are suitable interpreted.

The layout of Table 5 should be optimized with regard to line breaks.

Author Response

Thank you for your suggestion. We have revised Table 5 and Table 6 for optimizing the layout.